# Hypermethylation of SCAND3 and Myo1g Gene Are Potential Diagnostic Biomarkers for Hepatocellular Carcinoma

**DOI:** 10.3390/cancers12082332

**Published:** 2020-08-18

**Authors:** Fei Xu, Lulu Zhang, Yuxia Xu, Di Song, Wenting He, Xiaomeng Ji, Jianyong Shao

**Affiliations:** 1State Key Laboratory of Oncology in South China, Collaborative Innovation Center for Cancer Medicine, Sun Yat-Sen University Cancer Center, Guangzhou 510060, China; xufei1@sysucc.org.cn (F.X.); zhanglul@sysucc.org.cn (L.Z.); xuyx@sysucc.org.cn (Y.X.); songdi@sysucc.org.cn (D.S.); hewt@sysucc.org.cn (W.H.); Jixm@sysucc.org.cn (X.J.); 2Department of Molecular Diagnostics, Sun Yat-Sen University Cancer Center, Guangzhou 510060, China

**Keywords:** hepatocellular carcinoma, DNA methylation, diagnosis, biomarkers

## Abstract

Presently, there is a lack of effective blood-based biomarkers facilitating the diagnosis of hepatocellular carcinoma (HCC). Thus, we aimed to investigate novel methylation markers for HCC diagnosis, and explore relationships between biomarker methylation and clinicopathology of HCC. The methylation status of the SCAN domain containing three (SCAND3) and myosin 1g (Myo1g) genes in HCC cell lines and tissues were detected by digital droplet PCR. The serum SCAND3 and Myo1g methylation levels were analyzed in HCC-afflicted patients and unafflicted controls. The results indicated SCAND3 and Myo1g methylation were abnormally high in the HCC cell lines and tissues. The values of serum SCAND3, Myo1g, and SCAND3 + Myo1g methylation with respect to facilitating the detection, and early detection of HCC were better than for alpha-fetoprotein (AFP) alone. Furthermore, when we combined SCAND3 + Myo1g with AFP, a high sensitivity and specificity resulted. Notably, in the AFP-negative HCC group, the methylation of SCAND3 and Myo1g also showed an excellent diagnostic performance. Besides this, a high serum SCAND3 methylation level was an independent risk factor for predicting portal vein tumor thrombus (PVTT) in HCC patients (OR = 4.746, *p* = 0.013). Finally, SCAND3 and Myo1g enhanced the HCC diagnostics as noninvasive serum methylation biomarkers, and the SCAND3 methylation status effectively indicated HCC accompanied by PVTT.

## 1. Introduction

Globally, hepatocellular carcinoma (HCC) is the third leading cause of cancer-related deaths. Although strategies and outcomes for patients afflicted with HCC have improved recently, long-term prognoses remain bad, in part because of the lack of effective approaches for early diagnoses [1,2]. Therefore, improving the methods that can be used for the early detection of HCC is a key component for successful treatment and the improvement of survival rates. Typically, alpha-fetoprotein (AFP) and ultrasonography have been the main approaches to screen for HCC. However, the limitations of AFP and ultrasonography are well recognized and relate to the inadequate sensitivity and specificity for detecting HCC [3,4]. Previous research has also indicated that the AFP levels yielded modest measures of sensitivity (39–65%) and specificity (76–94%) [5]. In addition, the majority of patients with early-stage HCC are missed during screening based upon their AFP levels alone. Thus, due to unsatisfactory diagnostic accuracies, AFP measurements alone are not recommended by the American Association for the Study of Liver Diseases or by the European Association for the Study of the Liver for HCC Diagnosis [1,6]. Therefore, other methods, such as noninvasive assessments of HCC-related biomarkers, are necessary, especially for increasing the identification of early-stage HCC diagnoses.

DNA methylation is a universally recognized pattern in epigenetic modifications which regulates gene transcription and plays a key role in carcinogenesis [7,8]. Tumor suppressor gene hypermethylation is a contributor to transcriptional silencing and has been commonly observed in most types of human malignancies. Since tumor gene methylation is remarkably stable and can be assessed in different bodily fluids including the blood, it has emerged as a noninvasive molecular marker for cancer screening, prognostic prediction, and treatment-based evaluation [9,10,11]. Moreover, alterations in the levels of DNA methylation have been found to generally occur during the early phases of tumorigenesis and can be monitored dynamically. Thus, the monitoring of tumor gene methylation holds great potential for increasing the accuracy and precision of the diagnosis of early-stage tumors that might be present in HCC afflictions.

Recently, several studies have examined measures of circulating tumor DNA methylation, and their relative use as diagnostic or prognostic tools for the assessment of many types of cancers, including for HCC. The RAS association domain family 1A (RASSF1A) aberrant methylation in blood has been confirmed for use as a diagnostic and prognostic biomarker for HCC-afflicted patients [12,13,14,15,16,17]. Pasha Heba F et al. found that the sensitivity and specificity of RASSF1A methylation for HCC diagnoses were 40% and 86% respectively [13]. Other research indicated that long interspersed nucleotide element-1 (LINE-1) hypomethylation was an effective biomarker which facilitated the relatively early-stage diagnosis of HCC and predictions of its recurrence and unfavorable overall survival rates [12,18]. Tian et al. reported that the sensitivity of serum HCCS1 promotor methylation in detecting HCC was 62.5% [19]. Xu et al. found that by way of using the 10-gene CpGs methylation status, they could discriminate HCC patients from healthy controls with a relatively high sensitivity (>83%) and specificity (>90%) [20]. Recent findings from Zhang et al. indicated that a combination of six HCC-specific hypermethylation sites demonstrated high measures of sensitivity and specificity (92% and 98%, respectively) [21]. Although many DNA methylation biomarkers have been investigated, few assessments have been applied in clinical settings, mainly due to insufficient diagnostic accuracies and small sample sizes, as well as the relatively high costs of methylation-related analyses. Thus, more sensitive and less invasive biomarkers with good cost–performance ratios are highly desirable for use in the diagnosis of HCC and need to be confirmed based upon clinical studies.

SCAN domain containing 3 (SCAND3) is one of the SCAN family members and associated with the transcriptional regulation of gene expression. At present, a few studies have asserted that the SCAN N-terminus domain is involved in cell survival, metabolism, and differentiation [22,23]. However, the cellular roles and biological mechanisms assigned directly to SCAND3 remain unclear. Qiu et al. used pyrosequencing to quantify the CpGs methylation levels in HCC-afflicted samples and identified a three-CpG-based signature, including SCAND3. The three-CpG-based signature facilitated predictions of recurrence for early-stage HCC patients, thus providing new insights into the dynamics underlying HCC development and progression [24].

Myosins play essential roles in chromosomal and centrosomal amplification, as well as DNA microsatellite instability. Increasing evidence has also indicated that myosins are also involved in the dynamics underlying the formation and development of various types of cancers [25,26]. Jessica L. Ouderkirk-Pecone et al. identified that the overexpression of the myosin gene consequently promoted the progression of human breast cancer [27]. Some research has also indicated that measures of the expression of myosin can be used as prognostic factors for the assessment of colorectal cancer (CC) recurrence, and that they are associated with CC progression and metastasis development [28]. Myosin 1 G (Myo1g) is one member of the family of myosins, and previous research has indicated that myosin 1 G (Myo1g) regulates cell elasticity, cell migration, exocytosis, and endocytosis in B-lymphocytes [29,30]. However, the diagnostic values of SCAND3 and Myo1g methylation in HCC have not been reported previously. Thus, in this study we sought to determine levels of SCAND3 and Myo1g methylation in the tissue and serum of HCC patients and whether they might be useful as non-invasive biomarkers for the diagnosis and management of HCC.

## 2. Results

### 2.1. DNA Methylation Levels of SCAND3 and Myo1g in HCC Cell Lines and HCC Tissue Samples

In order to obtain quantitative data for the DNA methylation levels, we used digital droplet PCR (ddPCR) to facilitate the measurement of the methylation frequency of SCAND3 and Myo1g in HCC cell lines and tissues. We firstly detected the two gene methylation levels in seven HCC cell lines (HepG2, HepB3, MHCC97H, MHCC97L, SK-Hep-1, Huh7, PLC) and in the human normal hepatocyte cell line, LO2. The results indicated that the SCAND3 and Myo1g methylation ratios were significantly increased in HCC-afflicted cell lines compared to the normal hepatocyte cell line, and that these differences were significant (*p* < 0.05, Figure 1).

To assess the effects of the demethylating agent 5-Aza-2′-Deoxycytidine (5-AZC) upon the SCAND3 and Myo1g methylation levels, the MHCC97H and SK-Hep-1 cell lines were treated with four doses of 5-AZC (0.0, 2.5, 5.0, or 10.0 µM). The results revealed that there was a dose-dependent decrease in the methylation levels of SCAND3 and Myo1g in both MHCC97H and SK-Hep-1 cells (*p* < 0.05, Figure 2).

Next, we further assessed the two gene methylation levels in 20 pairs of HCC tissues and corresponding tumor adjacent tissues. The results indicated that the SCAND3 and Myo1g methylation ratios were significantly higher in HCC tissues than in adjacent noncancerous tissues (*p* < 0.05, Figure 3).

### 2.2. Serum DNA Methylation Levels of SCAND3 and Myo1g in Different Groups

To analyze the serum SCAND3 and Myo1g methylation levels in different groups, the methylation status of HCC patients, HBV (hepatitis B virus)-related liver cirrhosis (LC) patients, benign liver disease (BLD) cases, and healthy controls (HC) were assessed by a methylation-specific polymerase chain reaction (MSP). The SCAND3 and Myo1g methylation ratios among HCC patients were significantly higher than those of LC patients, BLD cases, and HC. There were no significant differences between the SCAND3 methylation levels in the LC patients, BLD cases, and HC. However, the Myo1g methylation levels in the LC patients were higher than those in the BLD cases and HC. When we combined the two methylation markers, the methylation ratios of the two-gene panel “SCAND3 + Myo1g” were significantly different for comparisons between the four groups and are as follows: 92.73% in HCC patients, 36.53% in LC patients, 12% in BLD cases, and 8.16% in HC, respectively. Similarly, the methylation levels in the LC patients were higher than those in the BLD cases and HC (Figure 4).

### 2.3. Diagnostic Values of AFP and Methylation Markers in HCC

To facilitate our investigations of the diagnostic values of SCAND3 and Myo1g in the whole HCC cohort, all the participants in the LC and BLD as well as HC groups were pooled together as controls for the analyses. Our results indicated that the SCAND3 methylation had a better performance than AFP for the diagnosis of HCC. The sensitivity, specificity, positive predictive value (PPV), negative predictive value (NPV), Kappa, and area under the curve (AUC) were respectively: 73.3%, 94.7%, 93.8%, 76.5%, 0.674 and 0.840 (0.794–0.886), 69.7%, 89.4%, 87.8%, 73%, and 0.585 and 0.796 (0.744–0.847) for AFP. The performance of the Myo1g methylation biomarker in differentiating the HCC cases from controls was also better than that of the AFP AUC: 0.818 (0.769–0.867) vs. 0.796 (0.744–0.847), with a sensitivity of 78.8%, specificity of 84.8%, PPV of 85%, NPV of 78.5%, and Kappa of 0.633. When we combined the two methylation markers, the results yielded a high sensitivity of 92.2% and specificity of 79.5%, with an AUC of 0.861 (0.816–0.906). More importantly, the combination of “SCAND3 + Myo1g” and AFP achieved a 97.6% sensitivity and 76.2% specificity with an AUC of 0.869 (0.825–0.912) (Table 1). The receiver operating characteristic (ROC) curve was conducted as shown in Figure 5a.

### 2.4. Diagnostic Values of AFP and Methylation Markers in Early HCC

In the early-stage HCC group, the diagnostic values of SCAND3 or Myo1g alone or SCAND3 + Myo1g were better than those of AFP. When SCAND3 methylation was used as a diagnostic marker, the sensitivity was 62%, the specificity was 94.7%, the Kappa value was 0.612, and the AUC was 0.784 (0.697–0.870). When Myo1g methylation was used as a diagnostic marker, the sensitivity was 64%, the specificity was 84.8%, the Kappa value was 0.472, and the AUC was 0.744 (0.658–0.830). SCAND3 and Myo1g were then combined to become a new marker for HCC diagnosis, and the results indicated that the sensitivity, specificity, PPV, NPV, and AUC were significantly improved. Moreover, the combined AFP + SCAND3 + Myo1g biomarker was found to have had a further increased diagnostic efficiency, with a sensitivity of 94%, specificity of 76.2%, AUC of 0.851 (0.794–0.908), and Kappa of 0.575 (Table 2). The ROC curve was conducted as shown in Figure 5b.

### 2.5. Diagnostic Values of Methylation Markers in AFP-Negative HCC Patients

Among the 165 HCC patients, 50 were AFP-negative. In this study, we further assessed the accuracy of SCAND3 and Myo1g as diagnostic markers for AFP-negative HCC patients. Methylation makers of SCAND3 or Myo1g alone or SCAND3 + Myo1g also indicated a good diagnostic performance. The AUC for SCAND3 was 0.804 (95% CI 0.720–0.887), and had a sensitivity of 66%, specificity of 94.7%, PPV of 80.5%, and NPV of 89.4%. The AUC for Myo1g was 0.794 (95% CI 0.715–0.872), and had a sensitivity of 74%, specificity of 84.8%, PPV of 61.7%, and NPV of 90.8%. More importantly, the combination of SCAND3 + Myo1g had the highest sensitivity (92%) and specificity (79.5%), and had an AUC of 0.857 (0.799–0.916) (Table 3). The ROC curve was conducted as shown in Figure 5c.

### 2.6. Association between Methylation Markers and HCC Clinicopathological Features

The measures of association between methylation markers and clinicopathological features in HCC patients were evaluated, including for gender, age, HBV copy number, tumor size, tumor number, BCLC (Barcelona Clinic Liver Cancer) staging, vascular invasion, and others. The hypermethylation of SCAND3 was significantly correlated with the tumor size (χ^2^ = 4.595, *p* = 0.032) and portal vein tumor thrombus (χ^2^ = 8.967, *p* = 0.003). In addition, the hypermethylation of Myo1g was significantly correlated with the tumor size (χ^2^ = 10.839, *p* = 0.001) and BCLC stage (χ^2^ = 10.277, *p* = 0.016). However, when combined with the SCAND3 and Myo1g, no correlation was found between SCAND3 + Myo1g hypermethylation and the clinicopathological characteristics of HCC-afflicted samples (Table 4). Next, we conducted univariate and multivariate analyses to facilitate further investigations of the relationships between the gene methylation levels and clinicopathological features. The results indicated that SCAND3 hypermethylation was independently associated with an increased risk of portal vein tumor thrombus (OR = 4.746, 95% CI 1.387–16.240, *p* = 0.013). Similarly, there was no correlation between the two-gene panel “SCAND3 + Myo1g” methylation level and the HCC clinicopathological features (Table 5). 

## 3. Discussion

Presently, effective methods are not yet widely available or perfected for HCC diagnosis. While some relatively new molecular biomarkers have been investigated for HCC detection in recent years, none have been used in structured examinations in the clinical setting [31,32,33], in part because their implementation could be costly, and because of this the diagnostic-related needs of professionals are unfulfilled. Thus, more powerful and cost-effective types of biomarkers for HCC diagnosis are urgently needed and need to be examined in rigorous clinical settings.

In the present study, we first examined and found that SCAND3 and Myo1g aberrant hypermethylation existed in HCC cell lines and tissues. It should be noted that only SCAND3 methylation was found to be associated with early-stage HCC recurrence, and Myo1g has been not investigated previously in respect to human cancers. Therefore, our findings will provide newfound insights into the dynamics underlying HCC development and progression.

Next, we systematically analyzed the serum SCAND3 and Myo1g methylation status in HCC-afflicted patients and unafflicted controls, including for LC, BLD, and HC. The results indicated that the serum SCAND3 and Myo1g methylation ratios were significantly higher in HCC-afflicted patients compared to the unafflicted controls. The ROC curve analyses indicated that SCAND3 and Myo1g achieved significantly better measures than AFP did for HCC-related diagnoses. Furthermore, when we combined “SCAND3 + Myo1g + AFP” to become a new marker, the resultant sensitivity improved to 97.6%, with an AUC of 0.869 (0.825–0.912). In order to facilitate the determination of whether or not SCAND3 and Myo1g methylation could be good potential markers for early HCC detection, we analyzed the two gene methylation statuses in early-stage HCC patients. The analyses revealed that both SCAND3 and Myo1g methylation yielded better diagnostic values than AFP did. To our surprise, “SCAND3 + Myo1g + AFP” had results that indicated an even more increased diagnostic efficiency for the identification of early HCC, with a corresponding sensitivity of 94% and an AUC of 0.851 (0.794–0.908). Moreover, the diagnostic accuracies of SCAND3 and Myo1g methylation in AFP-negative HCC patients were excellent, and “SCAND3 + Myo1g” showed the best sensitivity of 92%, with an AUC of 0.857 (0.799–0.916).

We also investigated the measures of association between hypermethylation markers and HCC clinicopathological features. Our findings indicated that only serum SCAND3 hypermethylation was independently associated with an increased risk of portal vein tumor thrombus (OR = 4.746, 95% CI 1.387–16.240, *p* = 0.013). A previous study conducted by Qiu et al. supported our findings, in that the authors found that SCAND3 hypermethylation could predict recurrence in early-stage HCC patients [24]. Thus, our results further suggested that the SCAND3 methylation levels may be associated with the determination of the prognosis of HCC patients.

Compared with previous research, this study and our novel approaches offer several strengths. First, most previous research lacked a full assessment of the gene methylation status in HCC cell lines and tissue samples [34,35,36]. We used ddPCR to evaluate the SCAND3 and Myo1g absolute quantification methylation levels in 7 HCC cell lines and 20 pairs of HCC tissues. Second, some previous studies only investigated the single gene methylation levels [32,33,34], which yielded moderate sensitivities and specificities. Some previous research used a panel of genes including 6 or 10 markers [20,21], which increased the testing costs and time unavoidably. Meanwhile, our study used a two-gene panel that not only enhanced the HCC diagnostic accuracy but also had a good cost–performance-related effectiveness. Third, compared to other studies, we collected a sufficient sample size of HCC-afflicted cases as well as integrated controls, including LC, BLD, and HC. These measures facilitated the identification of the two gene methylation levels, and confirmed their diagnostic values in relation to early-stage HCC and AFP-negative HCC patients. Essentially, no previous research has focused upon the AFP-negative HCC patient cohorts as we did; our findings provide critically needed information to improve the understanding of the dynamics underlying HCC-related diagnostics.

However, there are some limitations to our study. Firstly, due to short clinical follow-up periods, we could not evaluate the relationships between the hypermethylation markers and the treatment responses, recurrence, and prognoses of HCC patients. Secondly, our study indicated that the serum Myo1g methylation frequency in HBV-related liver cirrhosis patients was 32.69%, which is higher than we observed in the BLD cases and HC. Combined with the information available in previous research [37,38], we hypothesize that etiological factors such as hepatitis virus infection might lead to aberrant DNA methylation. However, this did not affect the accuracy of HCC diagnosis, even for our assessments of early-stage cohorts. Thirdly, our research found that only serum SCAND3 hypermethylation was an independent predictor of portal vein tumor thrombus. In order to confirm this result, a further study with an increasingly larger sample size also performed in a clinical setting would be necessary. Lastly, we have not validated the biological functions of the two hypermethylation markers in respect to the dynamics underlying HCC development. Thus, in the future we plan to investigate their biological mechanisms that could be involved in the carcinogenesis of HCC.

## 4. Materials and Methods

### 4.1. Cell Lines and Tissue Specimens

Human hepatocarcinoma cell lines (HepG2, HepB3, MHCC97H, MHCC97L, SK-Hep-1, Huh7, PLC) and human normal hepatocyte cell line (LO2) were purchased from the Institute of Cell Biology of the Chinese Academy of Sciences (Shanghai, China). Cells were cultured DMEM or RPMI-1640 containing 10% fetal bovine serum at a constant temperature of 37 °C in a humidified chamber with a constant atmosphere of 5% CO_2_. For the validation of the role of SCAND3 and Myo1g methylation, we used demethylating agent 5-Aza-2′-Deoxycytidine (Sigma, St. Louis, MO, USA) in both the MHCC97H and SK-Hep-1 cell lines. The cells were treated with four doses (0.0, 2.5, 5.0, or 10.0 µM) of 5-AZC, and the medium containing 5-AZC was changed every 24 h. After 72 h of incubation, the DNA methylation levels were analyzed. Twenty pairs of fresh histopathologically verified HCC tissues and tumor-adjacent tissues were collected from Sun Yat-sen University Cancer Center. The clinicopathological data and information for the 20 HCC patients who underwent hepatectomies are reported in Appendix A.

### 4.2. Study Population and Blood Sample Collection

We enrolled a total of 316 subjects, including 165 HCC patients and 151 controls, consisting of 52 patients with HBV-related liver cirrhosis, 50 patients with benign liver disease, and 49 healthy controls. All the participants were recruited from the Sun Yat-sen University Cancer Center between November 2018 and April 2019. The HCC patients were diagnosed based upon The American Association for the Study of Liver Disease (AASLD) practice guidelines for the management of hepatocellular carcinoma (updated version, 2010). The HCC stage was determined according to the guidelines in the Barcelona Clinic Liver Cancer (BCLC) system. Early-stage HCC is defined in patients presenting the BCLC 0-A stage. The tumor size was measured by computed tomography (CT) and/or ultrasound. The cut-off value for the serum AFP levels was 25 ng/mL. The baseline clinicopathological parameters for enrolled participants are summarized in Table 6. Peripheral blood was collected from the controls and HCC patients prior to any applications of anti-tumor treatments. Serum was separated immediately from the blood samples by way of centrifugation, and was then stored at −80 °C until used for DNA extraction. All the patients and controls were given and returned written informed consent, and all aspects of the study were approved by the Sun Yat-sen University Cancer Center (approval number: B2020-217).

### 4.3. DNA Isolation and Bisulfite Conversion

DNA from the tissues and cell lines was isolated using the QIAamp DNA Mini Kit (Qiagen, Hilden, Germany). Circulating free DNA (cfDNA) was extracted from 1 mL of serum using the QIAamp DNA Blood Mini Kit (Qiagen, Hilden, Germany) following all the manufacturer protocols. DNA bisulfite conversion was performed using the EZ DNA Methylation Kit (Zymo Research, Irvine, CA, USA) following all the manufacturer protocols. Finally, a total volume of 20 μL of modified DNA was obtained, which was used immediately for PCR or stored for later use at −20 °C.

### 4.4. Digital Droplet PCR and Methylation-Specific Polymerase Chain Reaction

The tissue and cell line DNA methylation analyses were performed using the QX200™ Droplet Digital™ PCR System (Bio-Rad). The ddPCR reactions were carried out in 20 µL volumes, using 5 µL bisulfite-converted DNA, 0.5 μL of each primer (10 µM), 0.3 µL of each probe (10 µM), 10 µL 2× ddPCR SuperMix (Bio-Rad), and 3.4 µL of nuclease-free water. The thermal-cycling conditions were 95 °C for 10 min (1 cycle), 94 °C for 30 s and 56 °C for 1 min (45 cycles), 98 °C for 10 min (1 cycle), and 12 °C hold.

The serum DNA methylation analyses were performed using a quantitative real-time methylation-specific PCR program in a 7500 Sequence detector (Perkin-Elmer Applied Biosystems, Foster City, CA, USA). Each PCR amplification was carried out in a 96-well plate composed of positive and negative controls, as well as samples and blanks of only nuclease-free water. The PCR cycling conditions were as follows: 95 °C for 5 min, 20 cycles at 95 °C for 15 s and 66 °C for 30 s, 30 cycles at 95 °C for 10 s and 58 °C for 31 s. Each reaction was performed in a total of 15 µL of PCR mixture, which contained 5 µL of bisulfite-converted DNA; 0.5 µL of each primer (10 µM); 0.1 µL of each probe (10 µM); 1.3 µL of nuclease-free water; and 7.5 µL of the master mix, which consisted of MSP DNA polymerase, 10 × MSP PCR buffer, and dNTPs. All the primer sequences for methylation analysis were as follows: SCAND3 (forward: 5′-GTTATAAATTGAGCGGTAAGATATTTGC-3′; reverse: 5′-CCTCGCCCAAACTACTCCG-3′), Myo1g (forward: 5′-GGGTAGAAGGTTATTCGTTGTGTATTTC-3′; reverse: 5′-CAATATACACAAAATACTTAACTCACGTCCT-3′), and GAPDH (forward: 5′-GTGGAGAGAAATTTGGGAGGTTAG-3′; reverse: 5′-CAACACAAACACATCCAACCTACA-3′). The details for the SCAND3 and Myo1g methylation data analysis are presented in the Appendix A.

### 4.5. Statistical Methods

Statistical analyses were performed using the IBM SPSS software (SPSS version 20.0, IBM, USA). Student’s *t*-tests or Mann–Whitney U-tests were used to facilitate the evaluation of the measures of correlation between two quantitative variables. Chi-squared tests or Fisher’s exact tests were used to facilitate comparisons of the differences in the serum gene methylation levels between different enrolled groups, as well as to facilitate the assessment of associations between the gene methylation status and the clinicopathological characteristics of HCC patients. To facilitate the determination of methylation-related biomarkers and AFP in the diagnosis of HCC, receiver operating characteristic curves were completed, and the AUC was reported with 95% CIs. We also used univariate and multivariate logistic regression analyses to facilitate investigations of the relationships between the methylation markers and the recorded clinical variables in HCC patients. In the univariate analyses, variables with *p* values < 0.05 were then further included for subsequent multivariate analyses. The sensitivity, specificity, PPV, NPV, accuracy, and Kappa value were calculated to facilitate the measurement of the diagnostic values of the methylation markers and AFP for HCC patients. For the combined methylation marker, patients with unmethylated SCAND3 and Myo1g were defined as negative. In contrast, patients which had at least one of either the two SCAND3 and Myo1g genes which displayed hypermethylation were defined as positive. A *p*-value of less than 0.05 was considered statistically significant.

## 5. Conclusions

In summary, our study demonstrated that the two novel genes SCAND3 and Myo1g hypermethylation have significant values for HCC diagnosis. Combined with AFP, our two-gene panel achieved a high sensitivity and specificity in diagnosing HCC, as well as for diagnosing patients with otherwise difficult-to-identify early-stage HCC. Even in the AFP-negative patients, this panel also showed excellent clinical application for HCC diagnosis. Furthermore, we observed that the SCAND3 methylation status was associated with portal vein tumor thrombus in HCC patients, which indicated its potential usefulness as a predictor for HCC progression. Ultimately, we hope that our research has helped to lay a foundation upon which the two emerging methylation genes we assessed can be further developed and used as effective, efficient, and robust biomarkers for HCC-related diagnoses and prognostic predictions.

## Figures and Tables

**Figure 1 cancers-12-02332-f001:**
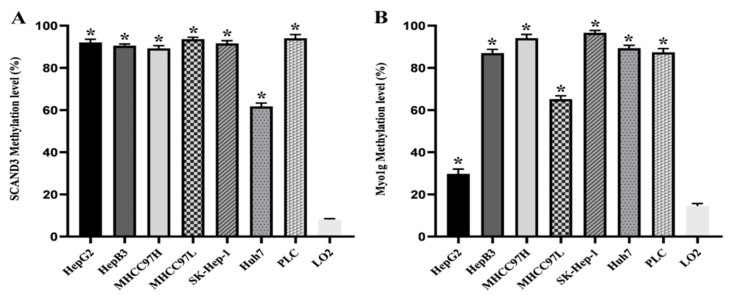
The methylation levels of SCAND3 and Myo1g in seven hepatocellular carcinoma (HCC) cell lines and the normal hepatocyte cell line LO2. (**A**) SCAND3 average methylation levels in seven HCC cell lines (HepG2, HepB3, MHCC97H, MHCC97L, SK-Hep-1, Huh7, PLC) and LO2 were 92.07%, 90.53%, 89.23%, 93.64%, 91.60%, 61.73%, 94.07%, 8.07%, respectively. (**B**) Myo1g average methylation levels in seven HCC cell lines (HepG2, HepB3, MHCC97H, MHCC97L, SK-Hep-1, Huh7, PLC) and LO2 were 29.73%, 87.00%, 94.10%, 65.17%, 96.96%, 89.30%, 87.37%, 14.47%, respectively. SCAND3 and Myo1g hypermethylation were more prominent in seven HCC cell lines than in LO2. Statistical comparisons were computed using the unpaired *t*-test, * *p* < 0.05.

**Figure 2 cancers-12-02332-f002:**
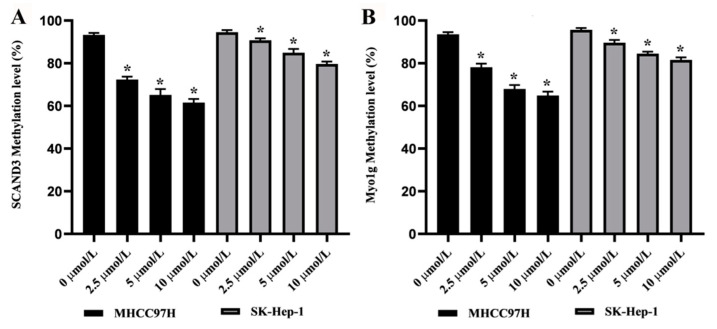
The methylation levels of SCAND3 and Myo1g in HCC cell lines treated with demethylating agent 5-Aza-2′-deoxycytidine (5-AZC). (**A**) After 0.0, 2.5, 5.0, or 10.0 µM of 5-AZC treatment, the average methylation levels of SCAND3 were 93.30%, 72.40%, 65.17%, and 61.57% in the MHCC97H cell line, and 94.57%, 90.76%, 84.93%, and 79.67% in the SK-Hep-1 cell line, respectively. (**B**) After 0.0, 2.5, 5.0, or 10.0 µM of 5-AZC treatment, the average methylation levels of Myo1g were 93.6%, 78.17%, 67.96%, and 64.90% in the MHCC97H cell line, and 95.70%, 89.67%, 84.50%, and 81.56% in the SK-Hep-1 cell line, respectively. Statistical comparisons were computed using the unpaired *t*-test, * *p* < 0.05.

**Figure 3 cancers-12-02332-f003:**
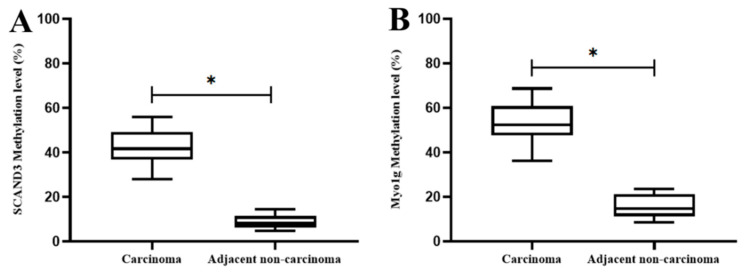
The methylation levels of SCAND3 and Myo1g in HCC tissues and adjacent non-cancerous tissues. (**A**) SCAND3 methylation ratios in cancerous tissues and adjacent noncancerous tissues were 41.79% and 8.73%, respectively; (**B**) Myo1g methylation ratios in cancerous tissues and adjacent noncancerous tissues were 53.43% and 15.67%, respectively. Statistical comparisons were computed using the unpaired *t*-test, * *p* < 0.05.

**Figure 4 cancers-12-02332-f004:**
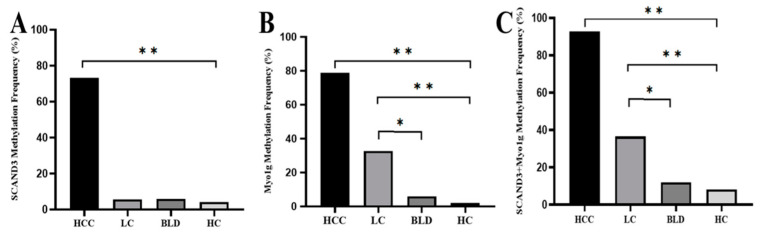
The methylation frequency of serum SCAND3, Myo1g, and SCAND3 + Myo1g in the HCC, HBV (hepatitis B virus)-related liver cirrhosis (LC), benign liver disease (BLD), and healthy controls (HC) groups. (**A**) The SCAND3 methylation ratio among HCC patients (73.33%) was significantly higher than that of LC patients (5.7%), BLD cases (6%), and HC (4.1%) (χ^2^ = 151.12, *p* < 0.0001). There were no significant differences between the SCAND3 methylation levels in LC patients, BLD cases, and HC (χ^2^ = 0.217, *p* = 0.897). (**B**) The Myo1g methylation ratio among the HCC patients (78.79%) was significantly higher than that of the LC patients (32.69%), BLD cases (6%), and HC (2%) (χ^2^ = 137.8, *p* < 0.001). The methylation levels in the LC patients were higher than that in the BLD cases (χ^2^ = 7.76, *p* < 0.05) and HC (χ^2^ = 16.18, *p* < 0.001). (**C**) The “SCAND3 + Myo1g” methylation ratio among the HCC patients (92.73%) was significantly higher than that of the LC patients (36.53%), BLD cases (12%), and HC (8%) respectively, (χ^2^ = 132.47, *p* < 0.001). The “SCAND3 + Myo1g” methylation levels in the LC patients were higher than those in the BLD cases (χ^2^ = 5.524, *p* < 0.05) and HC (χ^2^ = 11.55, *p* < 0.001). * *p* < 0.05, ** *p* < 0.01.

**Figure 5 cancers-12-02332-f005:**
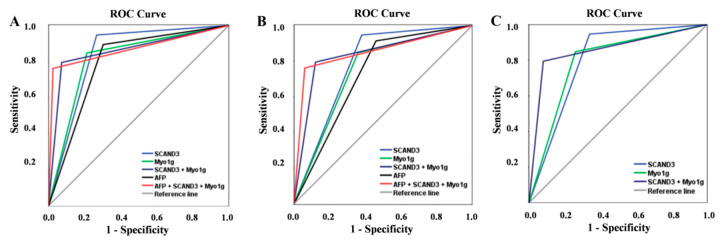
Diagnostic values of serum gene methylation makers and AFP for HCC patients. The receiver operating characteristic (ROC) of SCAND3, Myo1g, SCAND3 + Myo1g, AFP, and AFP + SCAND3 + Myo1g for HCC diagnosis in the whole HCC cohort (**A**), early-stage HCC patients (**B**), and AFP-negative HCC patients (**C**).

**Table 1 cancers-12-02332-t001:** Diagnostic values of serum gene methylation markers and AFP in predicting HCC.

Markers	Se (%)	Sp (%)	PPV (%)	NPV (%)	Accuracy (%)	AUC (95% CI)	Kappa
**AFP**	69.7	89.4	87.8	73.0	79.1	0.796 (0.744–0.847)	0.585
**SCAND3**	73.3	94.7	93.8	76.5	83.6	0.840 (0.794–0.886)	0.674
**Myo1g**	78.8	84.8	85.0	78.5	81.6	0.818 (0.769–0.867)	0.633
**SCAND3 + Myo1g**	92.2	79.5	83.2	90.2	86.2	0.861 (0.816–0.906)	0.720
**AFP + SCAND3 + Myo1g**	97.6	76.2	81.7	96.6	87.3	0.869 (0.825–0.912)	0.744

Se, sensitivity; Sp, specificity; PPV, positive predictive value; NPV, negative predictive value; AUC, area under the curve; CI, confidence interval; AFP, alpha-fetoprotein.

**Table 2 cancers-12-02332-t002:** Diagnostic values of serum gene methylation markers and AFP in predicting early-stage HCC.

Markers	Se (%)	Sp (%)	PPV (%)	NPV (%)	Accuracy (%)	AUC (95% CI)	Kappa
**AFP**	54	89.4	62.8	85.4	80.6	0.727 (0.636–0.818)	0.455
**SCAND3**	62	94.7	79.5	88.3	87.1	0.784 (0.697–0.870)	0.612
**Myo1g**	64	84.8	58.2	87.7	79.6	0.744 (0.658–0.830)	0.472
**SCAND3 + Myo1g**	88	79.5	58.7	95.2	81.6	0.837 (0.773–0.902)	0.578
**AFP + SCAND3 + Myo1g**	94	76.2	56.6	97.5	80.6	0.851 (0.794–0.908)	0.575

Se, sensitivity; Sp, specificity; PPV, positive predictive value; NPV, negative predictive value; AUC, area under the curve; CI, confidence interval; AFP, alpha-fetoprotein.

**Table 3 cancers-12-02332-t003:** Diagnostic values of SCAND3 and Myo1g methylation levels in AFP-negative HCC patients.

Markers	Se (%)	Sp (%)	PPV (%)	NPV (%)	Accuracy (%)	AUC (95% CI)	Kappa
**SCAND3**	66	94.7	80.5	89.4	87.5	0.804 (0.720–0.887)	0.646
**Myo1g**	74	84.8	61.7	90.8	82.1	0.794 (0.715–0.872)	0.551
**SCAND3 + Myo1g**	92	79.5	59.7	96.8	82.6	0.857 (0.799–0.916)	0.605

Se, sensitivity; Sp, specificity; PPV, positive predictive value; NPV, negative predictive value; AUC, area under the curve; CI, confidence interval; AFP, alpha-fetoprotein.

**Table 4 cancers-12-02332-t004:** Correlation between the methylation status and clinicopathological characteristics of the HCC patients.

Characteristic	N	SCAND3		*X^2^*	*P*	Myo1g		*X^2^*	*P*	SCAND3 + Myo1g		*X^2^*	*P*
		UM	M			UM	M			UM	M		
**Age (years)**													
**≤50**	73	20	53	0.036	0.850	16	57	0.039	0.843	4	69	0.624	0.429
**>50**	92	24	68			19	73			8	84		
**Sex**					0.498				0.183				
**Male**	147	38	109	0.459		29	118	1.776		11	136	0.088	0.766
**Female**	18	6	12			6	12			1	17		
**Tumor size (cm)**					0.032 *				0.001 *				
**≤5**	64	23	41	4.595		22	42	10.839		7	57	2.082	0.149
**>5**	101	21	80			13	88			5	96		
**Tumor number**													
**Single**	93	28	65	1.29	0.256	22	71	0.762	0.383	9	84	1.827	0.176
**multiple**	72	16	56			13	59			3	69		
**Lymphatic metastasis**													
**no**	146	42	107	2.860	0.091	34	112	3.268	0.071	11	135	0.129	0.720
**yes**	19	2	17			1	18			1	18		
**Distant metastasis**													
**no**	149	41	108	0.568	0.451	32	117	0.064	0.800	11	138	0.027	0.868
**yes**	16	3	13			3	13			1	15		
**BCLC stage**													
**0 + A**	50	19	31	6.760	0.080	18	32	10.277	0.016 *	6	44	3.177	0.365
**B + C**	115	25	90			17	98			6	109		
**Tumor differentiation**													
**Well-differentiated**	14	4	10	0.072	0.789	6	8	0.570		1	13	0.209	0.648
**Moderately/poorly differentiated**	62	20	42			20	42		0.450	7	55		
**Vascular invasion**													
**Microvascular invasion**													
**no**	51	14	37	1.233	0.269	19	32	0.638		4	47	1.185	0.276
**yes**	25	10	15			7	18		0.424	4	21		
**PVTT**													
**no**	122	40	82	8.967	0.003 *	29	93	1.833		11	111	2.111	0.146
**yes**	43	4	39			6	37		0.176	1	42		

N, number of patients; M, methylate status; UM, unmethylated status; PVTT, portal vein tumor thrombus; *, significant difference (*p* < 0.05).

**Table 5 cancers-12-02332-t005:** Multivariate logistic regression analysis of the clinicopathological characteristics of serum gene methylation markers in HCC.

Parameters	SCAND3				Myo1g				SCAND3+Myo1g			
	Univariate Analysis		Multivariate Analysis		Univariate Analysis		Multivariate Analysis		Univariate Analysis		Multivariate Analysis	
	OR (95% CI)	*P*	OR (95% CI)	*P*	OR (95% CI)	*P*	OR (95% CI)	*P*	OR (95% CI)	*P*		
**Age**	1.069 (0.534–2.139)	0.850			1.003 (0.971–1.035)	0.874			0.609 (0.176–2.107)	0.433		
**Sex (Male vs. Female)**	0.697 (0.245–1.987)	0.500			0.492 (0.170–1.420)	0.189			1.375 (0.167–11.322)	0.767		
**AST**	1.007 (0.999–1.014)	0.086			1.004 (0.997–1.011)	0.277			1.014 (0.994–1.035)	0.169		
**ALT**	1.006 (0.998–1.015)	0.158			1.001 (0.997–1.005)	0.674			1.006 (0.990–1.022)	0.465		
**ALB**	0.979 (0.904–1.060)	0.598			0.924 (0.843–1.013)	0.093			0.984 (0.859–1.127)	0.813		
**TBIL**	1.050 (0.997–1.107)	0.067			1.070 (1.004–1.139)	0.036	1.061 (0.996–1.131)	0.066	1.063 (0.961–1.175)	0.233		
**PT-INR**	0.460 (0.016–13.646)	0.653			0.443 (0.011–17.114)	0.663			0.406 (0.001–118.815)	0.756		
**Tumor size**	2.137 (1.060–4.309)	0.034	2.066 (0.750–5.687)	0.160	3.546 (1.629–7.719)	0.001	2.851 (0.958–8.481)	0.050	2.358 (0.175–7.778)	0.159		
**Tumor differentiation**	1.762 (0.642–4.834)	0.271			0.635 (0.194–2.076)	0.452			1.655 (0.187–14.647)	0.651		
**Tumor multiplicity**	1.508 (0.741–3.069)	0.357			1.406 (0.653–3.030)	0.384			2.464 (0.642–9.457)	0.189		
**BCLC staging**	1.540 (1.024–2.318)	0.038	0.838 (0.435–1.615)	0.597	1.948 (1.241–3.058)	0.004	1.182 (0.626–2.230)	0.607	1.687 (0.863–3.296)	0.126		
**Microvascular invasion**	1.190 (0.332–4.264)	0.789			0.655 (0.231–1.856)	0.426			2.238 (0.510–9.815)	0.285		
**PVTT**	4.756 (1.589–14.235)	0.005	4.74 (1.387–16.240)	0.013 ^*^	1.923 (0.738–5.012)	0.181			4.162 (0.521–33.239)	0.126		
**AFP**	1.000 (1.000–1.000)	0.166			1.000 (1.000–1.000)	0.184			1.000 (1.000–1.000)	0.471		
**HBV DNA (copies/mL)**	1.000 (1.000–1.000)	0.732			1.000 (1.000–1.000)	0.218			1.000 (1.000–1.000)	0.594		

AST, aspartate aminotransferase; ALT, alanine aminotransferase; ALB, albumin; TBIL, total bilirubin; PT-INR, prothrombin time-international normalized ratio; PVTT, portal vein tumor thrombus; *, significant difference (*p* < 0.05).

**Table 6 cancers-12-02332-t006:** Baseline characteristics of all the participants.

Variable	HCC (*n* = 165)	LC (*n* = 52)	BLD (*n* = 50)	HC (*n* = 49)
**Age (years)**	52.08 ± 11.74	51.58 ± 12.68	47.46 ± 14.67	39.57 ± 13.40
**Gender (male/female)**	147/18	42/10	32/18	21/28
**AST (U/L)**	51.80 (14.80–1193.90)	65.50 (12.00–317.00)	17.70 (11.50–1.80)	18.60 (11.60–42.10)
**ALT (U/L)**	43.40 (9.6–1344.7)	40.10 (4.00–205.00)	17.45 (5.00–47.40)	15.30 (7.00–107.00)
**TBIL (mg/dl)**	14.90 (4.80–78.30)	41.85 (7.10–618.00)	NA	NA
**ALB (g/L)**	42.20 (27.70–52.50)	37.55 (23.90–50.20)	NA	NA
**PT-INR**	1.05 (0.65–1.44)	NA	NA	NA
**HBV DNA (copies/mL)**	2127 (0–1.07E8)	1259 (0–2.32E8)	NA	NA
**AFP (ng/mL)**	225.3 (1.38–121.00)	25.86 (0.5–324.50)	4.99 (0.60–31.78)	3.45 (0.56–32.00)

Data are mean ± standard deviation or median (minimum-maximum) values. HCC, hepatocellular carcinoma; LC, liver cirrhosis; BLD, benign liver disease; HC, healthy control; AST, aspartate aminotransferase; ALT, alanine aminotransferase; TBIL, total bilirubin; ALB, albumin; PT-INR, prothrombin time-international normalized ratio; AFP, alpha-fetoprotein; NA, not available.

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
