# Peer review of "Hypermethylation of SCAND3 and Myo1g Gene Are Potential Diagnostic Biomarkers for Hepatocellular Carcinoma"

_cancers, 2020, doi:10.3390/cancers12082332_

Round 1

Reviewer 1 Report

The research manuscript by Xu et al investigated the hypermethylation status of SCAND3 and Myo1g gene as potential diagnostic markers for hepatocellular carcinoma. The manuscript is well written and seem interesting. The efforts taken by the author is appreciated. 

The authors however need to address some of the major concerns raised. 

1) DNA methylation levels in cell lines - The authors have tried to extensively investigate the hypermethylation status in 7 HCC cell lines. The negative control used in the study is obtained from healthy individuals. It is difficult to relate the two samples because there are so many biological parameters that is different. For a more accurate conclusions of the levels of hypermethylation, the authors need to use inhibitors of hypermethylation as reported in the HCC literatures. 

2) Diagnostic values - The authors have focused on the sensitivity of using SCAND3 or Myo1g alone or in combination. They need to substantiate how this diagnostic sensitivity is compared to the other widely available diagnostic markers for HCC. More detailed comparision is needed to substantiate the importance of SCAND3 and Myo1g.

Reviewer 2 Report

The authors have demonstrated utility of SCAND3 and Myo1g methylation as a potential diagnostic biomarker for HCC in liquid biopsy settings. Also, high methylation of SCAND3 in serum is correlated with PVTT. However I have several major issues with the study.

  1. Prevalence of methylation of SCAND3 and Myo1g in HCC should be reported. If this is already known then the information should be included in introduction otherwise available data of other HCC cohorts should be explored and presented as a figure. For eg. TCGA HCC cohort can be used for this purpose. 
  2. Vital details of methylation specific PCR are missing such as primer-probe sequences, region of the gene analysed and how many CpGs are covered ?
  3. Figure 1 and 2 legends should state the statistical tests performed on the data.
  4. Besides using blood cells as negative control, normal hepatocytes should also be used to determine the methylation level of the genes and compared with HCC cell lines. In terms of normal hepatocytes, immortalised human hepatocyte cell lines can be used.
  5. 20 HCC paired cases were used to prove hypermethylation of SCAND3 and Myo1g. It will be useful to know expression levels of the genes in the samples to determine correlation between gene expression and DNA methylation. Likewise, functional consequence of methylation of these genes should be explored in other existing HCC datasets.
  6. A important unanswered question of the study is: whether there is a difference in methylation level of SCAND3 and Myo1g between HBV-HCC versus non-HBV-HCC ? The serum cohort contains only HBV cases and hence it is essential to answer this question. 
  7. Details on how the combined methylation score is calculated should be included.
  8. Error bars in Fig 3 are missing.
  9. Fig 4 legend is too small. the figure will look better by unifying the colours for eg., Myo1g is green in a and b but black in c.
  10. Correlation between SCAND3 methylation in serum and PVTT is reported however there is no data to prove if this is just a correlation or is a functional consequence ? In the discussion, authors claim this to functional aspect of SCAND3 gene. As mentioned earlier, showing a correlation between gene methylation and expression can support the hypothesis.
  11. Introduction should be updated to include other studies reporting other methylation biomarkers for HCC detection such as RASSF1A and LINE1 elements. 

Round 2

Reviewer 1 Report

The efforts taken by the authors to revise the manuscript is appreciated. I recommend acceptance of the manuscript for publication. Please check for grammatical and other typo errors.

Reviewer 2 Report

Authors have addressed all the previous concerns and have made changes in the manuscript appropriately however, I still feel that the method and results regarding methylation quantification of the two genes should be described in more detail even if exact CpGs can't be discussed due to IP.

It is not mentioned how percentage methylation is calculated ? Now when primer details are added, it seems like GAPDH is used for normalisation however, there is no description relating to this.

Were there any standards used for  assay optimisation, if so please describe. 

For the cell lines and tissue samples data is shown as % methylation however, serum data is only presented as frequency of cases showing hypermethylation or not ? Again it will be useful for the readers, what were the values of % methylation in serum and what was the cut-off used to call the gene as hypermethylated in an individual. This data should be added at least as a supplemental information.
